# Cerebral White Matter Connectivity in Adolescent Idiopathic Scoliosis: A Diffusion Magnetic Resonance Imaging Study

**DOI:** 10.3390/children9071023

**Published:** 2022-07-10

**Authors:** David C. Noriega-Gonzalez, Jesús Crespo, Francisco Ardura, Juan Calabia-del Campo, Carlos Alberola-Lopez, Rodrigo de Luis-García, Alberto Caballero-García, Alfredo Córdova

**Affiliations:** 1Department of Surgery, Ophthalmology, Otorhinolaryngology and Physiotherapy, Faculty of Medicine, University of Valladolid, 47005 Valladolid, Spain; 2Traumatology and Orthopaedics Department, Hospital Clínico Universitario de Valladolid, 47005 Valladolid, Spain; jcrespos@saludcastillayleon.es (J.C.); fardura@saludcastillayleon.es (F.A.); 3Department of Radiology Hospital Clínico Universitario de Valladolid, 47005 Valladolid, Spain; juancalabia@gmail.com; 4ETSI Telecomunicación, Image Processing Laboratory. University of Valladolid, 47005 Valladolid, Spain; caralb@tel.uva.es (C.A.-L.); rodlui@tel.uva.es (R.d.L.-G.); 5Department of Anatomy and Radiology, Health Sciences Faculty, GIR: “Physical Exercise and Aging”, University of Valladolid, Campus Universitario “Los Pajaritos”, 42004 Soria, Spain; alberto.caballero@uva.es; 6Department of Biochemistry, Molecular Biology and Physiology, Health Sciences Faculty, GIR: “Physical Exercise and Aging”, University of Valladolid, Campus Universitario “Los Pajaritos”, 42004 Soria, Spain

**Keywords:** adolescent idiopathic scoliosis, magnetic resonance imaging, fractional anisotropy, connectomics techniques, MRI, AIS, MRtrix, tractography

## Abstract

Adolescent idiopathic scoliosis (AIS) is characterized by the radiographic presence of a frontal plane curve, with a magnitude greater than 10° (Cobb technique). Diffusion MRI can be employed to assess the cerebral white matter. The aim of this study was to analyze, by means of MRI, the presence of any alteration in the connectivity of cerebral white matter in AIS patients. In this study, 22 patients with AIS participated. The imaging protocol consisted in T1 and diffusion-weighted acquisitions. Based on the information from one of the diffusion acquisitions, a whole brain tractography was performed with the MRtrix tool. Tractography is a method to deduce the trajectory of fiber bundles through the white matter based on the diffusion MRI data. By combining cortical segmentation with tractography, a connectivity matrix of size 84 × 84 was constructed using FA (fractional anisotropy), and the number of streamlines as connectomics metrics. The results obtained support the hypothesis that alterations in cerebral white matter connectivity in patients with adolescent idiopathic scoliosis (AIS) exist. We consider that the application of diffusion MRI, together with transcranial magnetic stimulation neurophysiologically, is useful to search the etiology of AIS.

## 1. Introduction

Adolescent idiopathic scoliosis (AIS) is a complex deformity of the spine in all three dimensions of space. AIS is a high incidence alteration in the pediatric population (approximately 4% of the global schoolchildren population) [1,2,3,4]. It is believed that AIS is a multifactorial disorder, whose etiology and mechanism have not yet been fully elucidated [1,2,3,4]. AIS is characterized by the radiographic presence of a frontal plane curve, the magnitude of which is greater than 10° (Cobb technique) [3,4]. Structural scoliosis is characterized by vertebral and trunk rotation [5]. It is also interesting to use the Lenke classification system, which is a simple, accurate and reproducible way of classifying scoliosis [6]. Despite an increasing interest from the scientific community in AIS, its causes and pathophysiology are not yet fully understood. Moreover, despite numerous theories or hypotheses investigating the pathogenesis of AIS, new findings are constantly emerging [6].

AIS is accompanied by different risk factors, such as growth disturbances, postural disorders, environmental factors, visual and dental disorders, etc. [7,8,9]. AIS may also be related to high-risk sports (ballet, rhythmic gymnastics, swimming, some athletic activities, etc.) that may lead to altered spinal biomechanics [10].

Magnetic resonance imaging (MRI) has been used to further understand AIS. Diffusion MRI (dDMRI) is a method of signal contrast generation based on the differences in Brownian motion. dMRI can be employed to analyze the microarchitecture of the human body, especially in the white matter of the brain [11]. In this regard, for example, Shi et al. [1] performed segmentation of the vestibular system from high-resolution anatomical T2 images, and they observed significant differences between patients and healthy controls in certain geometric features of the left vestibular system. More recently, Chu et al. [12] and Kong et al. [13], using dMRI, found alterations in the connectivity of the medulla oblongata and in several intervertebral segments of the spinal cord (C1-2, C2-3, C3-4 and C4-5). Other authors [14] found decreases in fractional anisotropy, a white matter descriptor based on dMRI, in part of the corpus callosum in patients, compared to healthy controls, although this study had serious endocrinological drawbacks.

Although the main application of dMRI in clinical routine is the early diagnosis of ischemic stroke [13,14,15], it can also be applied to many other neurological disorders. One of the main features of this method is that the complex structural organization of the brain’s white matter can be shown in great detail [16,17,18,19].

Tractography [16] is a procedure used to reveal neural tracts, representing them by lines in the three-dimensional space in which the resonance acquisition is located. In other words, trajectories of the fiber bundles can be estimated in the brain and represent them. Thanks to tractography and a segmentation of the cerebral cortex into different regions of interest, we can perform a study of connectomics. Connectomics obtains measures of connectivity between different regions of the brain, including the number of fibers between two regions or the mean AF. From the resulting data, a connectivity matrix can be created to represent them, depending on a series of parameters that characterize them, obtaining a different matrix for each characteristic.

In AIS, there may be neurophysiological dysfunction and neuromorphological changes associated to changes in the medulla and cerebrum, including low cerebellar tonsils, abnormalities in white matter density, cerebral cortical thickness and corpus callosum morphology [20,21,22,23,24]. With these grounds, it seems clear that the neural system has an important contributor to the etiopathogenesis of AIS.

In this study, we analyze, by means of diffusion MRI, if any alteration exists in the connectivity of the cerebral white matter. We also investigate if there are observable differences in the white matter microstructure of the brain of patients with adolescent idiopathic scoliosis (AIS).

## 2. Materials and Methods

Twenty-two patients with AIS, and eighteen healthy controls (control group) participated in the study. Participants were informed about the study and provided written consent, either personally or through their parents or guardians (in the case of minors) (Table 1). Patients were included in the control group among volunteers from the database of the Spine Dept. who had a consultancy related to back problems in the range of age of the case study, with a normal X-ray and physical exam. Basically, we divided them according to a threshold of 40°, as this is a standard value threshold to limit conservative versus surgical treatment. Our objective was to analyze patients in the earlier phase of their scoliosis to analyze the data and obtain conclusions that could help to minimize the evolution of the scoliosis

The inclusion criterion for patients was, exclusively, to have been diagnosed with adolescent idiopathic scoliosis with Cobb angular values between 15–40°. Angular values were taken into consideration for the manuscript according to the sample of patients that we had. It was not really a concern to include patients between 10–14°. It will be part of a future research strategy to evaluate the differences according to the Cobb angle. 

Exclusion criteria were having congenital vertebral or other organ malformations and suffering from metabolic disorders.

Patients with AIS had no clinical symptoms because it is a disorder of posture in space. They did not have clinical data as all were asymptomatic. However, the clinical examination included an assessment of tone, motor strength, sensation and limb reflexes. An examination of abdominal reflexes and a detection of nystagmus was also performed, as described by other authors [25]. The neurological examination was performed by a pediatric orthopedic (neuropediatrician) and spine surgeon with expertise in pediatric examinations and spinal deformities.

The study complied with the ethical standards of the Declaration of Helsinki and was approved by the Ethics Committee of the Hospital Clínico Universitario de Valladolid-Área de Salud Valladolid Este (Spain) (Data: 19 May 2016) (Ref. EPA-10-33).

The acquisition protocol employed included an anatomical T1-weighted acquisition and a diffusion acquisition. The parameters of the first sequence were: a turbo field echo (TFE) sequence, a matrix size of 256 × 256, a spatial resolution of 1 × 1 × 1 mm^3^ and 160 sagittal slices covering the whole brain. The diffusion acquisition parameters were: 61 gradient directions plus a baseline volume without diffusion weighting, b-value = 1000 s/mm^2^, 128 × 128 matrix size, 2 × 2 × 2 mm^3^ spatial resolution and 66 axial slices covering the whole brain. The total duration of the acquisition was 19 min.

### 2.1. Magnetic Resonance Imaging (MRI)

It is essential that the surgeon and radiologist work together to ensure early recognition of treatable causes of scoliosis, to define the need (or not) and the timing of surgical intervention. In principle, the visible alterations of the vertebral mechanics are easy to appreciate physically and radiologically; however, a deeper and more detailed study of the alterations in the connectivity of the cerebral white matter requires much more advanced techniques such as MRI.

Imaging was performed with a Philips Achieva 3T MRI unit (Philips Healthcare, Best, The Netherlands). The MRI acquisition included one anatomical T1-type image and two diffusion acquisitions. Diffusion MRI is especially designed for detecting and measuring molecular diffusion in vivo, i.e., the translation of molecules.

For the anatomical acquisitions, the protocol consisted in a turbo field echo acquisition, with a voxel size of 0.865 × 0.875 × 1 mm and a 320 × 320 matrix with 170 sagittal slices covering the whole brain and cervical area (Figure 1).

In the first diffusion acquisition, a spin echo sequence was performed, using 32 gradient directions, b = 1000 s/mm^2^, a voxel size of 1.66 × 1.66 × 2 mm and a 144 × 144 matrix with 140 axial slices covering the whole brain and cervical area (Figure 2).

Finally, the second diffusion acquisition, centered on the cervical area, also used a spin echo sequence, using b = 1000 s/mm^2^, a voxel size of 1.7 × 1.7 × 2 mm and a 128 × 128 matrix with 24 coronal slices covering the whole cervical area and reaching part of the brain including the corpus callosum, the pons and the medulla oblongata (Figure 3).

In addition, cardiac synchronization was performed using a VitaGuard^®^ VG 2100 cardiorespiratory monitor (GETEMED; 14513 Teltow, Germany) for patient monitoring and to avoid pulsating artefacts.

Details of the parameters employed in the MRI acquisitions are summarized in Table 2.

#### 2.1.1. Image Processing

From the anatomical (T1) images, nonbrain structures were removed using a tool from FSL (http://fsl.fmrib.ox.ac.uk, accessed on 31 December 2021). Then, cortical segmentation was performed using the Freesurfer tool (https://surfer.nmr.mgh.harvard.edu/, accessed on 31 December 2021). Based on the information from the first diffusion acquisition, a streamline tractography of the whole brain was performed with the MRtrix tool (https://www.mrtrix.org/, accessed on 31 December 2021) using 2 million streamlines for each subject. Tractography [14] is an algorithmic method to deduce the trajectory of fiber bundles through the white matter based on dMRI data (Figure 4). Finally, by combining cortical segmentation with tractography, a connectivity matrix of size 84 × 84 was constructed using FA (fractional anisotropy) and the number of streamlines as metrics of connectomics. FA is a scalar value between zero and one that describes the degree of anisotropy of a diffusion process. FA provides information about the degree of diffusion anisotropy and is a measure of white matter integrity widely used in dMRI. An FA value of 0 means that diffusion is fully isotropic and the value 1 means that diffusion is purely linear [16,19].

#### 2.1.2. Connectomics

An important application within neuroscience is connectomics, the study and production of connectomes [17]. Connectomes [26] are a set of connections in a neural network. They describe the networks of synaptic connections between neurons in the brain. To study them, different regions of interest in the brain structure are defined as the nodes of the network, and the fiber tract connections between them are obtained from tractography [17]. From the resulting data, a connectivity matrix representing them can be created.

### 2.2. Statistical Analysis

The connectomics matrix provided a total of 3570 connections. However, many of them were very weak connections, with very few streamlines connecting certain cortical regions. Since the analysis of these connections can be prone to errors and noise, only those connections with at least 500 streamlines for all subjects were considered for further analysis. Thus, from the initial 3570 connections, 159 exceeded the threshold described above and were subjected to a statistical analysis. The statistical analysis was carried out using the programming language system for statistical computing and statistical graphics R (r-project.org). For this statistical analysis, a general linear model (GLM) was used, studying the influence of age, gender and patient or healthy control status on the connectivity values obtained.

First, a pooled analysis of all the connections that exceeded the threshold (global connectivity) was carried out, followed by an individual analysis of the connections. In the latter case, a correction for multiple comparisons was used, controlling the FDR (false discovery rate). An ANCOVA was performed with categorical variables (sex and cases/controls) and age as a continuous predictor. In the scenario where the coefficients that accompany the covariates are nonzero, then their influence is relevant. The covariates, in this case, produce a change in the mean. Adjusting all together also takes into account the influence of the covariates on each other.

## 3. Results

Analyzing connectivity globally (Table 3) and using FA as a metric of this connectivity, it was observed that there were statistically significant differences between the connectivity of patients with scoliosis and those of the control group, having adjusted for age and sex.

Regarding the individual analysis of the connections, Table 3 shows those in which statistically significant differences were found, out of the 159 connections analyzed. After correction for multiple comparisons, statistically significant differences were found in three connections considering the FA obtained:(a)Caudal-middle-frontal cortex (left hemisphere) with superior-frontal cortex (left hemisphere);(b)Connection of the isthmus of the left cingulate gyrus with itself;(c)Connection of the right cerebellum cortex with itself.

In five other connections, we found significant differences considering the adjusted average FA along these connections (Table 2):(a)Left hemispheric pericalcarine cortex with itself;(b)Left superior frontal cortex with left putamen;(c)Right superior frontal cortex with itself;(d)Insula (deep temporal lobe) with itself;(e)Superoparietal cortex with itself.

In all cases, FA increases were found in the patient group with respect to the control group.

We performed the analysis with different group values below 20°, 20–40° and above 40, and we did not find any significant results, probably because of the sample size. Future studies are planned with a bigger sample size to try to obtain some clarification

Considering the number of tractography streamlines as a metric, and also applying correction for multiple comparisons, significant differences were found in one connection: supramarginal cortex of the right hemisphere with itself.

## 4. Discussion

First of all, we must point out that in our study we carried out an exploratory analysis, namely, we did not start from any a priori hypothesis about which connections we suspected may be altered. For this reason, a correction for multiple comparisons was required, because we were analyzing a large number of comparisons and if we did not do so, we could find false positives produced simply by chance. This is why it was difficult to find significant differences, because the statistical threshold we applied was very strict. However, we believe that our observations can contribute to establishing diagnostic parameters to support the etiology of AIS.

We have in mind the findings of other authors [27,28] that suggest that the site of SSEP (Abnormal somatosensory evoked potential) abnormality in AIS is originated at a level above the cervical spine and it is likely to be related to alterations along the spinocortical pathway.

Diffusion MRI connectomics assesses anatomical rather than functional connectivity, namely, the connections quality of the pathways, not the amount of stimulus passing through them [17,18,19,29]. The most important finding of this study was the ability to analyze neuroanatomical pathways in scoliotic patients and compare them with healthy controls.

Therefore, the results obtained in this study support the hypothesis of the existence of alterations in the connectivity of the cerebral white matter in patients with AIS. We did not find decreases in FA in the regions mentioned in previous studies by other authors [11,13].

The pathways affected by increases in FA were the motor areas of the frontal cortex and the cingulate gyrus, which is an area of great structural and functional complexity. The cerebellar cortex connections are sensory pathways, although they are influential in modulating the extrapyramidal pathway. In all these cases, the interpretations of the data suggest that this increase in FA is a response to overstimuli, or positive feedback. The involvement of these pathways would affect body posture, balance and the degree of movement necessary to achieve the voluntary goal. In this way of exploring cortical connections in subjects with AIS, Xue et al. [30] observed a corresponding reduction of FA in fibers interconnecting primary somatosensory cortex and visual cortex. In addition, they also observed a relatively reduced FA on the left side in AIS patients with a right scoliotic curve.

In our work we also found the supramarginal gyrus to be affected, corresponding to Brodmann’s area 40, which is located in the parietal region (sensory area). However, we do not have an explanation for this anatomical finding; this could be a good starting point for future studies.

As for the alteration of the connections of the structures with themselves, they are interpreted as variations in the self-regulation of these centers in relation to compensation mechanisms in the pathway involved, in this case, the motor pathway.

In general, we tend to associate pathology with hypofunction or loss of function (weakness, paresis, hyporeflexia) but in some systems (such as motor/posture/tone) inhibitory functions are as important as excitatory ones. In these cases, the pathology may be due to hyperfunction (abnormal movements, tremor, spasticity, hyperreflexia) or a predominance of excitatory over inhibitory functions which could be an explanation for the increased connectivity of some pathways. This fact would also explain the increase in global connectivity in patients with scoliosis compared to controls. This hypothesis would open up an avenue for studying the pathophysiology of scoliosis, a possible imbalance between the excitatory and inhibitory pathways that modulate tone, posture and coordinated movement.

### Imitations of the Study

As for the limitations of the study, the most relevant we have identified are the number of streamlines generated by our whole brain tractography approach, which could be small, the connections, the number of subjects and the gender mismatch between groups. However, even though the number of streamlines employed in the whole brain tractography may be low, we believe that this problem is not critical since fiber density was taken into account in the tractography algorithm that was employed. Moreover, this study also has the limitation that, given the situation and the result of the analysis, it is difficult to indicate which fibers correspond to motor function. However, this is not of great significance, as AIS does not necessarily have to be caused by motor dysfunction.

All in all, even though our study is preliminary, we reported interesting findings, but additional studies are required in order to corroborate the existence of white matter alterations in AIS patients.

In conclusion, and in view of the data obtained, we can consider the routine application in the future of diffusion MRI together with neurophysiological studies of transcranial magnetic stimulation to continue the search for the etiology of AIS. It is logical that connectomics techniques may contribute to understanding the neurological influence on the pathophysiology of idiopathic scoliosis, although it will be necessary to combine anatomical and functional connectomics in larger groups of patients and controls.

## Figures and Tables

**Figure 1 children-09-01023-f001:**
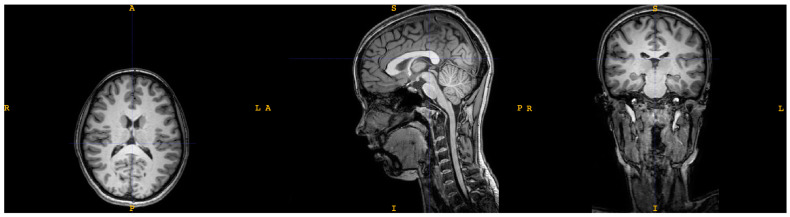
Example of T1 anatomical acquisition (axial, sagittal and coronal slices).

**Figure 2 children-09-01023-f002:**
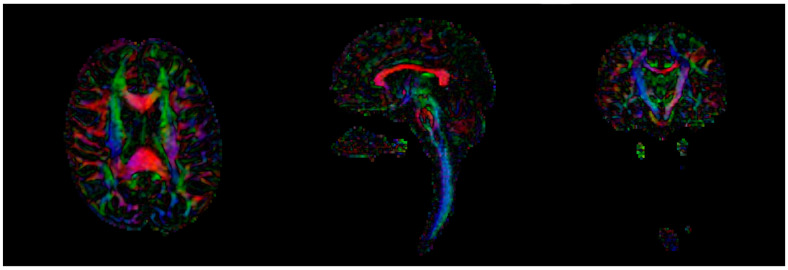
Color-coded FA in an example of the first diffusion acquisition (axial, sagittal and coronal slices).

**Figure 3 children-09-01023-f003:**
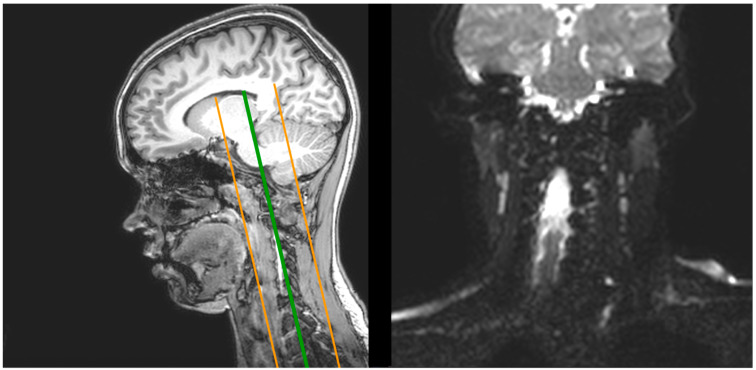
(**Left**) Over a sagittal view of the T1 anatomical slice, illustration of the coverage of the second diffusion MRI acquisition. (Right) Sample coronal view of the second diffusion MRI acquisition.

**Figure 4 children-09-01023-f004:**
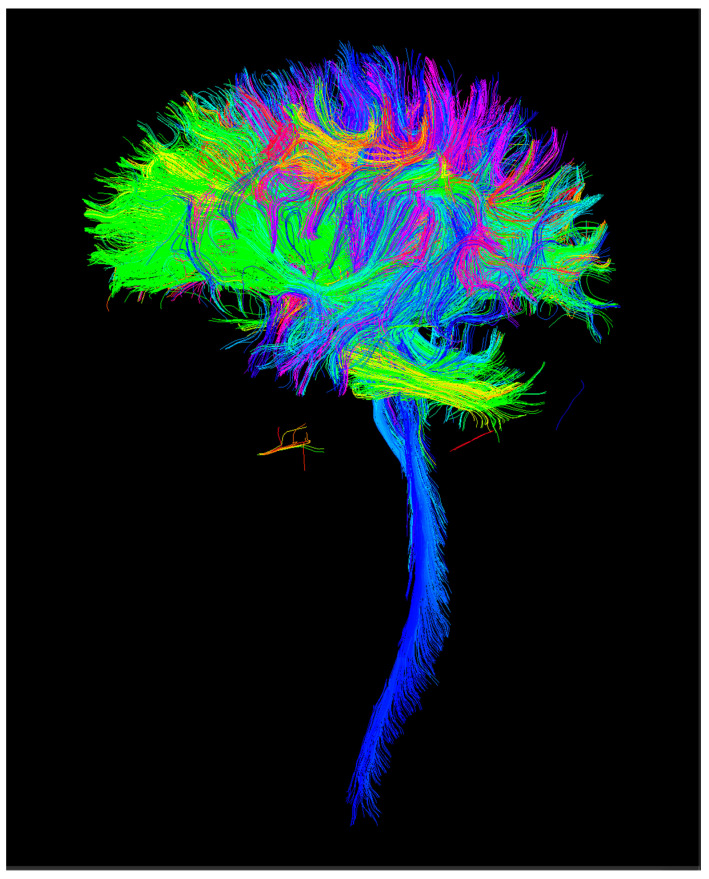
Sample case of whole brain tractography from the diffusion MRI data.

**Table 1 children-09-01023-t001:** Demographic details of the patient participant pool.

	*n*	Sex and Female/Male Ratio	Age (X ± SD)
Patients	22	17/5	14.73 ± 3.03
Controls	18	8/10	12.33 ± 2.43

**Table 2 children-09-01023-t002:** Details of the MRI acquisitions employed.

	T1	dMRI
Sequence type	Turbo field echo	Diffusion-weighted single shot spin echo
Repetition time	8.1 ms	9000 ms
Echo time	3.7 ms	86 ms
Flip angle	8°	90°
Echo train length	170	59
No. of slices	240	140
B-value	-	1000 s/mm^2^
No. of gradient directions	-	61
Orientation	Sagittal	Axial
Acquisition duration	359 s	696 s

**Table 3 children-09-01023-t003:** Connections in which statistically significant differences were found comparing the sample of patients with scoliosis with the control group.

Connection	Parameter	*p*-Value
Global connectivity	FA	3.8·10^−6^
Caudal-middle-frontal cortex (left hemisphere) to superior-frontal cortex (left hemisphere)	FA	<0.0092
Connection of the isthmus of the left cingulate gyrus with itself	FA	<0.0092
Connection of the right cerebellum cortex with itself	FA	<0.0092
Left hemispheric pericalcarine cortex to itself	FA (adjusted average)	0.003806
Left superior frontal superior cortex with left putamen	FA (adjusted average)	0.005745
Right superior frontal superior cortex with itself	FA (adjusted average)	0.008644
Insula (deep temporal lobe) with itself	FA (adjusted average)	0.008463
Superoparietal cortex with itself	FA (adjusted average)	0.003526
Supramarginal cortex	Number of tractography lines	<0.0062

## Data Availability

All data could be found in Hospital Clínica Universitario de Valladolid.

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
