# Peer review of "Cerebral White Matter Connectivity in Adolescent Idiopathic Scoliosis: A Diffusion Magnetic Resonance Imaging Study"

_children, 2022, doi:10.3390/children9071023_

Round 1

Reviewer 1 Report

Interesting and inspiring work clearly and correctly presented data. The weakness of the article is the large difference between the Cobb angle among the respondents. It is worth dividing patients into groups with a low angle, e.g., 10-25 degrees, and groups with a large angle, e.g., above 40. This would allow distinguishing the CNS changes from etiological to secondary ones resulting from many years of advanced disease.

Author Response

REV-1

Dear Reviewer, we thank you for the opportunity to revise our manuscript and we appreciate the careful review and constructive suggestions.

Interesting and inspiring work clearly and correctly presented data. The weakness of the article is the large difference between the Cobb angle among the respondents. It is worth dividing patients into groups with a low angle, e.g., 10-25 degrees 25-40, and groups with a large angle, e.g., above 40. This would allow distinguishing the CNS changes from etiological to secondary ones resulting from many years of advanced disease.

Lines: 100-103

Basically we divided them into below 40 as this is a standard value threshold to limit conservative versus surgical treatment. Our objective was to analyse patients in the earlier phase of their scoliosis to analyse the data and obtain conclusions that could help to minimize the evolution of the scoliosis

Lines: 241-245

We performed the analysis with different group values below 20º, 20º-40º and above 40 and we did not find any significant results. Probably because of the sample size. Future studies are planned with bigger sample size to try to obtain some clarification

Reviewer 2 Report

Idiopathic scoliosis is a serious orthopedic problem. Cause-and-effect treatment is limited due to its unknown etiology. Each attempt to find the cause of idiopathic scoliosis deserves a positive reception from the world of science.

The description of the research methodology describes in detail the inclusion and exclusion criteria for the study group, but there is no information about the control group, ie patients who did not have idiopathic scoliosis. Was X-ray imaging diagnostics, which eliminates scoliosis in a patient, a criterion for inclusion in this group? It is not clear on what basis patients without scoliosis were included in the control group. The angular range of the Cobb angle of 15-40 degrees is also worth considering, why such values? Why were patients with 10-14 degrees scoliosis excluded?

The conducted research is as up-to-date and interesting as possible, and minor comments on the research methodology do not reduce its quality. I hope that in the future the authors will continue their research on a larger group of patients.

Author Response

REV-2

Dear Reviewer, we thank you for the opportunity to revise our manuscript and we appreciate the careful review and constructive suggestions.

Idiopathic scoliosis is a serious orthopedic problem. Cause-and-effect treatment is limited due to its unknown etiology. Each attempt to find the cause of idiopathic scoliosis deserves a positive reception from the world of science.

The description of the research methodology describes in detail the inclusion and exclusion criteria for the study group, but there is no information about the control group, ie patients who did not have idiopathic scoliosis.

Lines: 98-100

Patients were included in the control group among volunteers from the data base of the Spine Dept. that had a consultancy related to back problems in the range of age of the case study with a normal X-ray and physical exam.

- Was X-ray imaging diagnostics, which eliminates scoliosis in a patient, a criterion for inclusion in this group? It is not clear on what basis patients without scoliosis were included in the control group.

The normal X-ray was an inclusion criterion. This would be answered in the preceding lines (95-98).

- The angular range of the Cobb angle of 15-40 degrees is also worth considering, why such values?

We did not have any patients in the case study group below 15º, of course we could modify it to 10-40º, but it would not have change the sample.

- Why were patients with 10-14 degrees scoliosis excluded?

Lines: 108-111

Angular values were taken into consideration for the manuscript according to the sample of patients that we had. Not really a concern to include patients between 10-14º. It is part of the future research strategy to evaluate differences according to the Cobb angle

The conducted research is as up-to-date and interesting as possible, and minor comments on the research methodology do not reduce its quality. I hope that in the future the authors will continue their research on a larger group of patients.

This manuscript is a resubmission of an earlier submission. The following is a list of the peer review reports and author responses from that submission.

Round 1

Reviewer 1 Report

Thanks for your responses. I do appreciate that reanalysis would take more time. However, if it is unfeasible to perform re-analysis, do provide your justification and explain this limitation in your discussion. Specifically this work is limited by the lack of accounting for fibre density which confounds the streamline counts. Secondly, in your discussion, please explain that the regions you found do not correspond to motor function - and suggest some reasons why - one of these reasons should be the methodological limitation. I see no reason why connectivity matrices could not be shown, or why relationship to clinical data could not be shown. If there is a real reason why not, again, please state it in the discussion section.

Author Response

REV-1

Thanks for your responses. I do appreciate that reanalysis would take more time. However, if it is unfeasible to perform re-analysis, do provide your justification and explain this limitation in your discussion. Specifically, this work is limited by the lack of accounting for fibre density which confounds the streamline counts. Secondly, in your discussion, please explain that the regions you found do not correspond to motor function - and suggest some reasons why - one of these reasons should be the methodological limitation. I see no reason why connectivity matrices could not be shown, or why relationship to clinical data could not be shown. If there is a real reason why not, again, please state it in the discussion section.

Thank you very much for your comments which will help to improve the work. Please find below the answers and where we have placed the suggested issues

Limitations have been introduced at the end of the discussion as a separate heading.

Limitations of the study

As for the limitations of the study, it has several. Among them are mainly the number of flow lines analysed, which could be small, the connections, the number of subjects, the strict gender match between groups and the gender mismatch between groups.

In this regard, we should mention that, although the data and checkpoints allow us to assess the situation, we would like to point out that these are preliminary data given the novelty of the technology applied to this pathology. It is currently not possible to carry out a new analysis, as this would mean carrying out a new study given the large number of hours required by the system for its analysis. This leads to a lower specificity of fibre density, which may confound the counting of the current lines. On the other hand, the study also has the limitation that, given the situation and the result of the analysis, it is difficult to indicate which fibres correspond to motor function. However, this is not of great significance, as AIS does not necessarily have to be caused by motor dysfunction.

Patients with AIS have do no present clinical symptoms because is a disorder of posture in space. They don't have clinical data in our study group as all are asymptomatic. However, the clinical examination included assessment of tone, motor strength, sensation and limb reflexes. An examination of abdominal reflexes and detection of nystagmus was also per-formed, as described by other authors (25). The neurological examination was performed by a paediatric orthopaedic and/or spine surgeon with expertise in paediatric examinations and spinal deformities.

Respect the connectivity matrices, they are not shown, only those with a certain number of streamlines and which I have indicated are shown.

Reviewer 2 Report

The manuscript entitled "Cerebral White Matter Connectivity In Adolescent Idiopathic Scoliosis. A Diffusion and Connectomic Resonance Imaging Study" is original and stimulates future studies and reflections.

1) I advise the authors to argue the introduction by including known risk factors. You can consult the following articles: 

Scaturro D, Costantino C, Terrana P, Vitagliani F, Falco V, Cuntrera D, Sannasardo CE, Vitale F, Letizia Mauro G. Risk Factors, Lifestyle and Prevention among Adolescents with Idiopathic Juvenile Scoliosis: A Cross Sectional Study in Eleven First-Grade Secondary Schools of Palermo Province, Italy. Int J Environ Res Public Health. 2021 Nov 24;18(23):12335. doi: 10.3390/ijerph182312335. PMID: 34886069; PMCID: PMC8656498.

2) The patient group and the control group do not seem uniform. I advise you to compare age and male/female ratio.

3) the materials and methods, especially the choice of scans should be better specified.

4) The discussion is well articulated. I find that one of the limits is also not having the clinical data of the 22 patients (degree of scoliosis, muscular problems, postural problems ...)

Author Response

REV-2

The manuscript entitled "Cerebral White Matter Connectivity In Adolescent Idiopathic Scoliosis. A Diffusion and Connectomic Resonance Imaging Study" is original and stimulates future studies and reflections.

Thank you very much for your comments which will help to improve the work. Please find below the answers and where we have placed the suggested issues

1) I advise the authors to argue the introduction by including known risk factors. You can consult the following articles:

AIS is accompanied by different risk factors, such as growth disturbances, postural disorders, environmental factors, visual and dental disorders, etc. [7,8,9]. AIS may also be related to high-risk sports (ballet, rhythmic gymnastics, swimming, some athletic ac-tivities, etc.) that may lead to altered spinal biomechanics [10].

  1. Kim, S.; Uhm, J.Y.; Chae, D.H.; Park, Y. Low body mass index for early screening of adolescent idiopathic scoliosis: A comparison based on standardized body mass index classifications. Asian Nurs. Res. 2020, 14, 24–29.
  2. Pan, X.X.; Huang, C.A.; Lin, J.L.; Zhang, Z.J.; Shi, Y.F.; Chen, B.D.; Zhang, H.W.; Dai, Z.Y.; Yu, X.P.;Wang, X.Y. Prevalence of the thoracic scoliosis in children and adolescents candidates for strabismus surgery: Results from a 1935-patient cross-sectional study in China. Eur. Spine J. 2020, 29, 786–793.
  3. Scaturro D, Costantino C, Terrana P, et al. Risk Factors, Lifestyle and Prevention among Adolescents with Idiopathic Juvenile Scoliosis: A Cross Sectional Study in Eleven First-Grade Secondary Schools of Palermo Province, Italy. Int J Environ Res Public Health. 2021;18(23):12335. Published 2021 Nov 24. doi:10.3390/ijerph182312335
  4. Latalski, M.; Danielewicz-Bromberek, A.; Fatyga, M.; Latalska, M.; Kröber, M.; Zwolak, P. Current insights into the aetiology of adolescent idiopathic scoliosis. Arch. Orthop. Trauma Surg. 2017, 137, 1327–1333.

2) The patient group and the control group do not seem uniform. I advise you to compare age and male/female ratio.

The male/female ratio is different because the incidence of cases is 10/1 higher than in the general population. When looking groups of adolescent volunteers it is difficult to standardise by age and it has not been possible to find healthy controls from earlier stages of adolescence. The problem is that even though it is a non-invasive test we did not find candidates with parental consent.

3) the materials and methods, especially the choice of scans should be better specified.

In material and methods we have introduced the following paragraph

It is essential that the surgeon and radiologist work together to ensure early recognition of treatable causes of scoliosis, to define the need or not, and the timing of surgical intervention. In principle, the visible alterations of the vertebral mechanics are easy to appreciate physically and radiologically, however, the deeper and more detailed study of the alterations in the connectivity of the cerebral white matter requires much more advanced techniques such as MRI.

4) The discussion is well articulated. I find that one of the limits is also not having the clinical data of the 22 patients (degree of scoliosis, muscular problems, postural problems ...)

Patients with AIS have no present clinical symptoms because is a disorder of posture in space. They don't have clinical data as all are asymptomatic. However, the clinical examination included assessment of tone, motor strength, sensation and limb reflexes. An examination of abdominal reflexes and detection of nystagmus was also per-formed, as described by other authors (25). The neurological examination was performed by a paediatric orthopaedic and/or spine surgeon with expertise in paediatric examinations and spinal deformities.

Round 2

Reviewer 1 Report

Thanks for adding the limitations. Please again check the Figure legends for Spanish text and English grammar throughout. 

Author Response

Thank you very much for your clarifications. We have corrected the foot of figures. Also we have had them corrected by an English teacher. Anyway, if you need to improve your English, we will request it to the publisher.